# Culturally adapted quasi-experimental interventions for reducing entomophobia and disgust: A study among older adults in Iran and Malaysia

entomophobia; disgust; older adults; quasieExperimental; culturally adapted interventions; Iran; Malaysia; visual and textual instruments

**Corresponding author:**
Abdolrahim Asadollahi;
Email: a.asadollahi@hotmail.co.uk

Aboozar Soltani[1], Mahsa Nasrabadi[2], Siti Anom Binti Ahmad[3] and Abdolrahim Asadollahi[4,5,6] 

[1]Research Center for Health Sciences, Institute of Health, Department of Medical Entomology and Vector Control, School of Health, Shiraz University of Medical Sciences, Iran, Islamic Republic of; [2]Department of Gerontology, Shiraz University of Medical Sciences, Iran, Islamic Republic of; [3]Malaysian Institute of Ageing, Universiti Putra Malaysia, Malaysia; [4]Director of Research Centre for Youth Population and Active Aging (RCYPAA), Shiraz University of Medical Sciences, Zand Ave., Shiraz, Iaran; [5]Head of Department of Gerontology, School of Health, Shiraz University of Medical Sciences, Razi Ave., Shiraz, Iran and [6]ACQoL, Deakin University, Australia

## Abstract

This study investigated factors influencing insect phobia among older adults in Iran and Malaysia using a quasi-experimental design with individual and group-based teaching models. The study included 151 older adults (82 Iranians, 69 Malaysians). Baseline and post-intervention scores were analyzed using paired *t*-tests, MANOVA, regression analysis, path analysis and a neural network model. Malaysians scored higher on EVI: Disgusting Pre (mean = 8.03 vs. Iranian mean = 7.46, $p < 0.05$) but showed greater reductions post-intervention (mean difference = 1.03 vs. Iranian mean difference = 0.63, $p < 0.01$). OAEAS scores decreased more among Malaysians (mean difference = 14.45 vs. Iranian mean difference = 15.08, $p < 0.05$). Males reported higher fear and disgust levels than females ($p < 0.05$). Pet ownership reduced phobic responses ($p < 0.05$), while chronic conditions heightened baseline scores but limited reductions over time. Group-based interventions were more effective for Malaysians, while individual-based approaches worked better for Iranians. The neural network model explained 82% of EVI variance and 79% of OAEAS variance. Culturally tailored interventions effectively reduce insect phobia among older adults. Future research should explore longitudinal effects and broader cultural contexts.

## Impact statement

This study shows that culturally adapted interventions are the most effective in reducing Entomophobia and disgust. Increased knowledge about insect pathogenicity unexpectedly intensified these reactions. Group-based interventions benefited Malaysian older adults, while individualized approaches worked better for Iranians. Pet ownership appeared protective, and combining visual and textual tools enabled a more comprehensive assessment of fear and disgust toward insects.

## Highlights

- Culturally adapted Interventions more effective in reducing entomophobia and disgust.
- Increased knowledge about insect pathogenicity paradoxically intensified fear and disgust responses.
- Group-based interventions more effective for Malaysian older adults, while individual approaches worked better for Iranians.
- Pet ownership acted as a protective factor.
- Visual and textual instruments provide comprehensive assessment of entomophobia and disgust.pt

## Introduction

Global advancements in public health have significantly increased life expectancy and improved the quality of life for older adults (Guzel et al., 2021). However, this growing aging population faces unique mental health challenges, including phobias such as Entomophobia and Arachnophobia. These intense fears of insects and arachnids disproportionately affect women (Kostuch, 2022) and can trigger physiological symptoms like dizziness and palpitations, impairing daily functioning (Zsido et al., 2023). Such fears often stem from early-life experiences or exaggerated perceptions of risk (Ruiz-García and Valero-Aguayo, 2023).

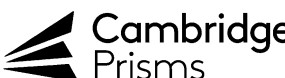



Medical entomology highlights the critical interactions between humans and insects, particularly in biodiverse regions. Arthropods, which encompass over 5.5 million species, are vectors for diseases such as malaria and arboviruses (Omuse et al., 2024). These factors exacerbate fears in tropical climates, like Malaysia and semi-arid regions, such as Iran. Despite its arid environment, Iran ranks high in arthropod diversity (Dicks et al., 2024), while Malaysia's tropical climate increases insect encounters, potentially intensifying Entomophobia.

Anxiety disorders affect 31.2% of individuals over their lifetime, with specific phobias accounting for 12.5% of cases (Kostuch, 2022). Entomophobia and Arachnophobia impact approximately 5% of the general population, with severe cases affecting 1% (Fukano and Soga, 2023). These phobias are influenced by gender, trauma and knowledge about diseases (Azil et al., 2021). Educational interventions, such as gamification, show promise in reducing irrational fears (Lampropoulos and Kinshuk, 2024).

This study examines fear and disgust related to arthropods among older adults in Iran and Malaysia, assessing individual and group teaching models using the Entomophobia Visual Instrument (EVI) and the Older Adults' Entomophobia and Arachnophobia Scale (OAEAS). The objectives include: (a) comparing pre- and post-intervention fear and disgust levels; (b) identifying predictors such as gender, pet ownership and chronic conditions; (c) evaluating cultural preferences in intervention design (EVI vs. OAEAS). Given the growing aging population globally, understanding and addressing phobias such as Entomophobia is crucial for improving mental health outcomes. Older adults face unique challenges due to chronic conditions, reduced mobility and environmental exposure, making culturally tailored interventions essential. Furthermore, comparing Iran and Malaysia provides valuable insights into how cultural and environmental factors mediate responses to behavioral interventions, contributing to the broader field of geriatric psychiatry.

Grounded in cultural psychology (Gong et al., 2020), we posit that fear and disgust are shaped by cultural beliefs and environmental exposure. For instance, older Malaysians' frequent arthropod contact may heighten visual sensitivity through repeated exposure, while Iranians' responses may rely more on cognitive appraisal influenced by their arid environment with fewer insect encounters (Burr et al., 2021). Cultural mechanisms such as social learning and cognitive framing play a critical role in shaping fear responses. For instance, older Malaysians' frequent arthropod contact may heighten visual sensitivity through repeated exposure, while Iranians' responses may rely more on cognitive appraisal influenced by their arid environment with fewer insect encounters. Understanding these mechanisms is vital for designing culturally adapted interventions that effectively reduce Entomophobia. Using path analysis, MANOVA, and neural networks, we explore how sociocultural factors – such as collectivism and environmental familiarity – mediate intervention effectiveness. Findings aim to guide culturally tailored strategies for managing Entomophobia (Tanega et al., 2025).

## Materials and methods

### Study design and population

This study used a quasi-experimental pre-post design based on cultural psychology to explore Entomophobia and Arachnophobia among older adults in Iran and Malaysia. The interventions, both individual and group-based, were adapted to align with cultural norms, experiences and environmental factors. Using two validated tools (EVI and OAEAS), fear and disgust levels were measured before and after the intervention.

The inclusion of Iran and Malaysia was intentional to capture both ecological and cultural contrasts relevant to Entomophobia. Ecologically, Iran represents a semi-arid climate with fewer arthropod encounters, while Malaysia's tropical environment provides constant exposure to diverse insect species. Culturally, the two nations also differ in their dominant social orientations – Malaysia aligns with collectivist values emphasizing social cohesion and shared learning, whereas Iran exhibits relatively more individualist characteristics that prioritize personal coping and introspection. This dual selection allows examination of how ecological context (climatic exposure) and cultural orientation (collectivism vs. individualism) jointly shape phobic and disgust responses among older adults.

The following sections outline the study population, sampling, tools, protocols and analyses:

### Study population and sampling procedures

The study used a two-stage stratified random sampling method to recruit adults aged 60 and older from urban centers in Iran and Malaysia. In the first stage, stratification was based on major urban districts to ensure geographical representativeness within each country. For Iran, three urban strata (Shiraz, Isfahan and Tehran) were defined according to the national health system's district structure. For Malaysia, participants were stratified into three major urban areas (Kuala Lumpur, Selangor and Penang) using the MyAgeing® registry framework. In the second stage, within each stratum, eligible older adults were randomly selected from local health or aging databases, ensuring proportional representation by gender and ethnicity (Malay, Chinese and Indian for Malaysian participants). This approach minimized sampling bias and enhanced comparability between national cohorts.

Iranian participants were selected via the national health system, while Malaysian participants were drawn from the MyAgeing® database. The sample size of 151 was calculated using PASS software (version 15), based on effect sizes from Nasrabadi et al. (2024), with a 12% margin of error, 95% confidence level and 5% dropout rate. Eligible participants were at least 60 years old, free of severe cognitive impairment and provided informed consent. Participants were coded in Excel and randomly selected using a computer-generated table, stratified by country and ethnicity for Malaysians (Malay, Chinese and Indian). The final sample included 82 Iranians (54.3%) and 69 Malaysians (45.7%), comprising 30 Malay (19.9%), 20 Chinese (13.2%) and 19 Indian (12.6%) individuals. Ages ranged from 60 to 85, with most (92.1%, $n = 139$) aged 60–75. Females made up 68.9% ($n = 104$) and males 31.1% ($n = 47$). Ethics approval was obtained, and all participants provided written consent.

### Measurement tools

The study examined demographic, behavioral and health-related variables, including nationality, gender, pet ownership, chronic conditions, knowledge about arthropod pathogenicity, teaching models and history of insect/arachnid bites, and two primary instruments were used to assess arthropod phobia and disgust:

- **Entomophobia Visual Instrument (EVI):** A visual-based tool consisting of 10 images of arthropods (locusts, beetles, butterflies, dragonflies, spiders, bees, ants, millipedes, ticks and flies).

Participants rated their fear and disgust levels for each image on a 3-point scale (0 = none, 1 = below 50% and 2 = 50% and above). In this scale, the terms "below 50%" and "50% and above" refer to the participant's subjective estimation of the intensity of their emotional response (fear or disgust) to each image, rather than to the likelihood of a behavioral reaction. The 50% threshold was introduced to help older adults categorize their feelings in relation to a moderate emotional intensity reference point, where values below 50% indicate mild discomfort and 50% or above reflect strong aversive reactions. This simplified structure was validated in pilot testing to improve comprehension and reduce cognitive load among participants compared to traditional 5-point formats. Total scores ranged from 0 to 20, with higher scores indicating greater fear or disgust. EVI was administered individually to ensure accurate responses.

- **Older Adults' Entomophobia and Arachnophobia Scale (OAEAS):** A self-reported Likert-scale questionnaire with 20 items, scored on a 5-point scale (1 = strongly disagree and 5 = strongly agree). Total scores ranged from 20 to 100, where higher scores reflected more severe phobic reactions. OAEAS was designed to capture cognitive appraisals of fear and disgust toward arthropods (Nasrabadi et al., 2024).

Baseline assessments were conducted prior to the intervention, followed by post-intervention evaluations after six weeks. Demographic information, such as age, gender, ethnicity, marital status, living arrangements, chronic conditions, pet ownership, knowledge about insect pathogenicity, and history of insect bites or stings, was collected through structured interviews.

### Intervention protocols

The intervention consisted of educational sessions tailored to cultural preferences:

- **Individual-Based Model:** Participants received one-on-one counseling sessions lasting approximately 30 min per session, focusing on personalized coping strategies, cognitive restructuring and exposure therapy. Sessions emphasized the biological characteristics of arthropods, their ecological roles, and ways to manage fear safely.
- **Group-Based Model:** Participants attended weekly group workshops lasting 60 min per session, facilitated by trained educators. These workshops incorporated interactive activities, discussions and collaborative learning approaches to normalize fear and promote social support. Group sizes ranged from 8 to 12 participants.

Both models incorporated balanced messaging about arthropod-related risks and benefits, avoiding excessive emphasis on pathogenicity to prevent amplifying fear. Participants who had been bitten by insects/arachnids or reported high baseline scores were given additional guidance during the intervention.

The educational interventions in Iran and Malaysia were delivered by separate teams of trained local educators to maintain linguistic and cultural appropriateness. To ensure standardization, all educators underwent centralized online training using an English-language intervention manual developed by the principal investigators. The manual was translated and back-translated into Persian and Bahasa Malay to ensure conceptual equivalence. Training sessions covered instructional content, cultural adaptation procedures and ethical considerations. Intervention fidelity was maintained through adherence checklists, periodic supervision meetings and random observation of sessions by site coordinators.

This approach ensured consistency in delivery across settings while respecting cultural nuances.

### Statistical analyses

Analyses were conducted using JAMOVI (version 2.6.25). Descriptive statistics summarized demographics and baseline data. Paired samples $t$-tests assessed pre- to post-intervention changes in EVI and OAEAS scores within subgroups, reporting effect sizes (Cohen's $d$, Glass's $\Delta$, Hedge's $g$). Independent samples $t$-tests compared mean differences by nationality, gender, pet ownership and chronic conditions. One-way ANOVA evaluated ethnic and categorical differences, with Tukey post-hoc tests. Repeated measures ANOVA examined time × nationality interactions on EVI/OAEAS scores, reporting partial eta-squared ($\eta^2$). MANOVA (Pillai's Trace) tested multivariate effects of predictors like nationality, gender, chronic conditions, pet ownership, arthropod knowledge, teaching models and bite/sting history. Regression models identified phobia outcome predictors, ensuring assumptions of linearity, independent errors (Durbin–Watson: 1.8–2.2), homoscedasticity, normal residuals (Shapiro–Wilk $p > 0.05$), no multicollinearity (VIF < 2) and no outliers (Cook's Distance <1). Adjusted $R^2$ and $f^2$ ensured robustness. Path analysis tested direct/indirect relationships, with fit indices: $\chi^2$ ($p$ = 0.13), CFI (0.94), TLI (0.92), RMSEA (0.05; 95% CI: 0.02–0.08) and SRMR (0.06). A feedforward neural network captured non-linear interactions, explaining 82% (EVI) and 79% (OAEAS) of variance.

### Environmental context and cultural adaptation

Guided by cultural psychology's culture-psychology interdependence framework, the study adapted instruments and teaching models to address culturally specific arthropod responses. The EVI and OAEAS were administered in both Iran and Malaysia for cross-cultural comparability, while individual- and group-based teaching models were implemented despite potential cultural differences in effectiveness. Environmental factors influenced instrument suitability: Malaysia's tropical climate and frequent arthropod exposure likely enhanced EVI visual stimulus responsiveness, whereas Iran's arid environment with fewer encounters favored OAEAS cognitive self-report measures. Intervention models aligned with cultural values – group-based approaches suited Malaysia's collectivist emphasis on social support, while individual counseling matched Iran's preference for personalized care, though relative effectiveness remains unclear. This dual-nation design balanced evaluation of instruments and models while respecting environmental and cultural contexts, providing empirical evidence on how these factors mediate outcomes and offering insights for culturally adapted phobia interventions.

Based on previous cross-cultural research, we hypothesized differential intervention efficacy across the two cultural contexts. Specifically, group-based interventions were expected to be more effective among Malaysian older adults, consistent with Malaysia's collectivist orientation, emphasizing social harmony and shared emotional regulation. In contrast, individual-based interventions were hypothesized to produce greater benefits for Iranian participants, reflecting Iran's growing preference for individualized counseling and self-reflective coping in health and psychological domains. These expectations were derived from prior studies highlighting how cultural values moderate behavioral intervention outcomes (Gong et al., 2020; Joo and Liu, 2021).

### Data management and quality control

Data collection followed standardized procedures, with trained interviewers administering questionnaires and visual assessments. McDonald's omega coefficients ensured internal consistency (EVI: $\omega = 0.89$; OAEAS: $\omega = 0.91$). Missing data (< 5%) were handled using mean imputation. All assumptions for parametric tests were verified, and outliers were excluded if Cook's Distance exceeded 1.

### Ethical considerations

Throughout the study, ethical guidelines were strictly followed. Participants were informed about the objectives, procedures, and their rights, including confidentiality and the option to withdraw without consequences. Cultural sensitivity was prioritized in intervention design and data collection to align with local norms and respect participants' backgrounds. Ethical approval was obtained from the XXXXX Committee (XXXXX), and local health authorities granted necessary permissions. Informed consent was provided by all participants in Persian and Bahasa Malay after a detailed explanation of the study. Data confidentiality and anonymity were rigorously protected, and no harm was reported during the study.

### Results

### Participants' characteristics

The study included 151 older adults: 82 (54.3%) Iranian and 69 (45.7%) Malaysian participants. Most were female (68.9%, $N = 104$), with males comprising 31.1% ($N = 47$). The majority (92.1%, $N = 139$) were aged 60–75, while 7.9% ($N = 12$) were 75–90. Iranians had a mean age of 67.52 (SD = 5.89), and Malaysians 69.38 (SD = 6.31). Chronic conditions were reported by 74.8% ($N = 113$): 43.7% ($N = 66$) of Iranians and 31.1% ($N = 47$) of Malaysians. Depression was more common among Iranians (11.9%, $N = 18$) than Malaysians (4.6%, $N = 7$), while respiratory diseases were more prevalent among Malaysians (6.0%, $N = 9$) than Iranians (4.6%, $N = 7$). Sleeping patterns differed, with 12.6% ($N = 19$) of Iranians and Malaysians sleeping over 8 h nightly. Fear of insects varied: scorpions were feared more by Iranians (24.5%, $N = 37$) than Malaysians (13.9%, $N = 21$). Large flying insects were the most feared, slightly more so among Iranians (27.8%, $N = 42$) than Malaysians (17.9%, $N = 27$). Knowledge of insect pathogenicity was higher among Iranians (37.7%, $N = 57$) than Malaysians (28.5%, $N = 43$). Behavioral responses, like seeking safe places, were more common among Iranians (33.8%, $N = 51$) than Malaysians (30.5%, $N = 46$). These findings highlight demographic, health and attitudinal differences between Iranian and Malaysian older adults.

### Baseline scores

Baseline scores for the Entomophobia Visual Instrument (EVI) and the Insect/arachnid Phobia Questionnaire (OAEAS) revealed significant cross-cultural differences. Malaysian participants scored higher on the EVI: Disgusting Pre (mean = 8.03, SD = 2.56) compared to Iranians (mean = 7.46, SD = 2.88; $t(149) = -2.24$, $p < 0.05$). However, no significant difference was observed in OAEAS pre scores between Malaysians (mean = 72.03, SD = 15.86) and Iranians (mean = 69.25, SD = 16.97; $t(147) = -1.02$, $p > 0.05$). These findings suggest that the visual-based EVI may be more sensitive to cultural and environmental differences, such as the higher prevalence of

insects/arachnids in Malaysia. For detailed baseline scores, see Supplemental File 1.

### Intervention effects

According to Table 1, paired samples *t*-tests revealed significant reductions in both EVI and OAEAS scores post-intervention. For EVI: Disgusting, Malaysians showed a larger mean reduction (mean difference = 1.03, SE = 0.22, $t(68) = 4.68$, $p < 0.01$, Cohen's $d = 0.55$) compared to Iranians (mean difference = 0.63, SE = 0.27, $t(81) = 2.35$, $p < 0.05$, Cohen's $d = 0.26$). Similarly, OAEAS scores decreased significantly for both groups, with Malaysians showing a mean reduction of 14.45 points (SE = 2.25, $t(68) = 6.42$, $p < 0.01$, Cohen's $d = 0.76$) and Iranians showing a reduction of 15.08 points (SE = 1.89, $t(81) = 7.98$, $p < 0.01$, Cohen's $d = 0.86$).

OAEAS scores decreased significantly in both groups, with Iranians showing a slightly greater numerical reduction (mean difference = 15.08, SE = 1.89, $t(81) = 7.98$, $p < 0.01$, Cohen's $d = 0.86$) compared to Malaysians (mean difference = 14.45, SE = 2.25, $t(68) = 6.42$, $p < 0.01$, Cohen's $d = 0.76$). Although both reductions were substantial, the slightly larger decrease among Iranian participants suggests greater cognitive restructuring of fear-related beliefs following the individualized sessions.

Group-based teaching models were more effective for Malaysians, while individual-based models showed better outcomes for Iranians. For example, Malaysians in group-based models had a mean reduction of 16.44 points in OAEAS Post scores (SE = 1.70, $t(67) = 9.61$, $p < 0.01$, Cohen's $d = 1.00$), compared to 9.18 points for individual-based models (SE = 2.95, $t(39) = 3.11$, $p < 0.01$, Cohen's $d = 0.40$). For detailed paired *t*-test results, see Table 1.

While several comparisons yielded statistically significant results ($p < 0.05$), the corresponding effect sizes ranged from small to moderate (Cohen's $d = 0.26$–$0.76$), indicating varying levels of practical significance. For instance, the mean reduction in EVI: Disgusting scores was larger for Malaysians (mean difference = 1.03) than for Iranians (mean difference = 0.63), but the effect size ($d = 0.26$) suggests that this difference has modest clinical implications. Similarly, gender differences in phobic responses were consistent across both populations, with males reporting higher scores than females ($p < 0.05$), but the effect sizes ($d = 0.32$–$0.45$) highlight the need for cautious interpretation of these findings.

The more pronounced reductions observed in OAEAS compared to EVI scores indicate that the interventions may have primarily facilitated cognitive desensitization rather than emotional habituation. While EVI captures immediate affective reactions to visual stimuli, OAEAS assesses cognitive appraisals of threat and disgust. Therefore, improvements in OAEAS reflect changes in rational understanding and attitude reappraisal – consistent with educational and cognitive-behavioral intervention mechanisms rather than exposure-based emotional adaptation (Burr et al., 2021; Ruiz-García and Valero-Aguayo, 2023).

### Interpretation of regression analysis results

To better understand the predictors of insect phobia among older adults in Iran and Malaysia, regression analyses were conducted using both the EVI OAEAS. The findings presented in Tables 2 and 3 reveal that gender was a consistent predictor across both populations, with males reporting higher baseline fear and disgust levels compared to females. For instance, in EVI: Disgusting Pre, males scored significantly higher than females (mean = 8.23 vs. mean = 7.49, $\beta = 0.38$, $p < 0.01$). This pattern was observed for both pre-

**Table 1.** Paired samples *t*-test results for pre- and post-intervention scores across subgroups in Iranian and Malaysian older adults

| Subgroups | Measure | Mean pre | Mean post | MD | SE | *T*-value | DF | *p*-value | Cohen's *d* | Glass's Δ | Hedge's *g* |
|---|---|---|---|---|---|---|---|---|---|---|---|
| 1. Iranians vs. Malaysians | EVI: Disgusting | 7.46 / 8.03 | 6.83 / 7 | 0.63 / 1.03 | 0.27 / 0.22 | 2.35 / 4.68 | 81 / 68 | p < 0.05 / p < 0.01 | 0.26 / 0.55 | 0.27 / 0.59 | 0.26 / 0.55 |
| | EVI: Phobia | 7.49 / 6.91 | 6.83 / 5.88 | 0.66 / 1.03 | 0.27 / 0.32 | 2.44 / 3.21 | 81 / 68 | p < 0.05 / p < 0.01 | 0.27 / 0.38 | 0.28 / 0.41 | 0.27 / 0.38 |
| | OAEAS | 69.25 / 72.03 | 54.17 / 57.58 | 15.08 / 14.45 | 1.89 / 2.25 | 7.98 / 6.42 | 81 / 68 | p < 0.01 / p < 0.01 | 0.86 / 0.76 | 0.88 / 0.78 | 0.86 / 0.76 |
| 2. Iranian males vs. females | EVI: Disgusting | 8.32 / 7.15 | 7.41 / 6.62 | 0.91 / 0.53 | 0.42 / 0.34 | 2.17 / 1.56 | 21 / 59 | p < 0.05 / p < 0.05 | 0.44 / 0.20 | 0.45 / 0.21 | 0.44 / 0.20 |
| | EVI: Phobia | 7.32 / 7.55 | 6.14 / 7.08 | 1.18 / 0.47 | 0.61 / 0.30 | 1.93 / 1.57 | 21 / 59 | p < 0.05 / p < 0.05 | 0.41 / 0.20 | 0.42 / 0.21 | 0.41 / 0.20 |
| | OAEAS | 66.86 / 70.14 | 48.09 / 56.4 | 18.77 / 13.74 | 3.47 / 2.25 | 5.41 / 6.08 | 21 / 59 | p < 0.01 / p < 0.01 | 1.16 / 0.78 | 1.18 / 0.80 | 1.16 / 0.78 |
| 3. Malaysian males vs. females | EVI: Disgusting | 8.16 / 7.95 | 6.84 / 7.09 | 1.32 / 0.86 | 0.48 / 0.41 | 2.75 / 2.09 | 24 / 43 | p < 0.05 / p < 0.05 | 0.54 / 0.31 | 0.55 / 0.32 | 0.54 / 0.31 |
| | EVI: Phobia | 7.72 / 6.45 | 6.44 / 5.57 | 1.28 / 0.88 | 0.55 / 0.48 | 2.33 / 1.83 | 24 / 43 | p < 0.05 / p < 0.05 | 0.48 / 0.28 | 0.49 / 0.29 | 0.48 / 0.28 |
| | OAEAS | 68.08 / 74.27 | 58.76 / 56.91 | 9.32 / 17.36 | 3.95 / 2.47 | 2.36 / 7.02 | 24 / 43 | p < 0.05 / p < 0.01 | 0.48 / 1.06 | 0.49 / 1.08 | 0.48 / 1.06 |
| 4. Iranians with chronic conditions | EVI: Disgusting | 7.92 / 7.13 | 7.14 / 6.21 | 0.78 / 0.92 | 0.20 / 0.41 | 3.90 / 2.24 | 81 / 38 | p < 0.01 / p < 0.05 | 0.43 / 0.46 | 0.44 / 0.47 | 0.43 / 0.46 |
| | EVI: Phobia | 7.29 / 7.03 | 6.50 / 6.11 | 0.79 / 0.92 | 0.24 / 0.43 | 3.29 / 2.14 | 81 / 38 | p < 0.01 / p < 0.05 | 0.38 / 0.48 | 0.39 / 0.49 | 0.38 / 0.48 |
| | OAEAS | 68.88 / 75.39 | 52.68 / 64.79 | 16.20 / 10.6 | 1.61 / 2.71 | 10.06 / 3.91 | 81 / 38 | p < 0.01 / p < 0.01 | 1.11 / 0.74 | 1.13 / 0.76 | 1.11 / 0.74 |
| 5. Malaysians with chronic conditions | EVI: Disgusting | 7.13 / 8.07 | 6.21 / 7.05 | 0.92 / 1.02 | 0.54 / 0.30 | 1.70 / 3.40 | 68 / 38 | p < 0.05 / p < 0.01 | 0.20 / 0.43 | 0.21 / 0.44 | 0.20 / 0.43 |
| | EVI: Phobia | 7.03 / 7.45 | 6.11 / 6.48 | 0.92 / 0.97 | 0.48 / 0.36 | 1.92 / 2.69 | 68 / 38 | p < 0.05 / p < 0.05 | 0.23 / 0.32 | 0.24 / 0.33 | 0.23 / 0.32 |
| | OAEAS | 75.39 / 70.95 | 64.79 / 58.43 | 10.60 / 12.52 | 2.25 / 2.51 | 4.71 / 5 | 68 / 38 | p < 0.01 / p < 0.01 | 0.56 / 0.66 | 0.57 / 0.67 | 0.56 / 0.66 |
| 6. Iranians with pets | EVI: Disgusting | 7.06 / 7.46 | 6.63 / 6.83 | 0.43 / 0.63 | 0.46 / 0.27 | 0.93 / 2.33 | 51 / 82 | p < 0.05 / p < 0.05 | 0.13 / 0.26 | 0.14 / 0.27 | 0.13 / 0.26 |
| | EVI: Phobia | 6.85 / 7.49 | 6.23 / 6.83 | 0.62 / 0.66 | 0.43 / 0.29 | 1.44 / 2.27 | 51 / 82 | p < 0.05 / p < 0.05 | 0.20 / 0.27 | 0.21 / 0.28 | 0.20 / 0.27 |
| | OAEAS | 70.40 / 69.25 | 59.08 / 54.17 | 11.32 / 15.08 | 2.30 / 1.57 | 4.92 / 7.98 | 51 / 82 | p < 0.01 / p < 0.01 | 0.68 / 0.86 | 0.69 / 0.88 | 0.68 / 0.86 |
| 7. Malaysians with pets | EVI: Disgusting | 8.07 / 8.03 | 7.05 / 7.00 | 1.02 / 1.03 | 0.24 / 0.22 | 4.25 / 4.68 | 98 / 69 | p < 0.01 / p < 0.01 | 0.43 / 0.55 | 0.44 / 0.57 | 0.43 / 0.55 |
| | EVI: Phobia | 7.42 / 6.91 | 6.48 / 5.88 | 0.94 / 1.03 | 0.36 / 0.32 | 2.61 / 3.21 | 98 / 69 | p < 0.05 / p < 0.01 | 0.38 / 0.38 | 0.39 / 0.40 | 0.38 / 0.38 |
| | OAEAS | 70.59 / 72.03 | 53.97 / 57.58 | 16.62 / 14.45 | 1.73 / 2.25 | 9.61 / 6.42 | 98 / 69 | p < 0.01 / p < 0.01 | 0.98 / 0.76 | 0.99 / 0.78 | 0.98 / 0.76 |
| 8. Iranian individual-based vs. group-based | EVI: Disgusting | 7.85 / 7.17 | 7.05 / 6.68 | 0.80 / 0.49 | 0.35 / 0.28 | 2.29 / 1.75 | 59 / 80 | p < 0.05 / p < 0.05 | 0.29 / 0.19 | 0.30 / 0.20 | 0.29 / 0.19 |
| | EVI: Phobia | 7.17 / 7.26 | 6.30 / 6.42 | 0.87 / 0.84 | 0.37 / 0.30 | 2.35 / 2.8 | 59 / 80 | p < 0.05 / p < 0.01 | 0.31 / 0.31 | 0.32 / 0.32 | 0.31 / 0.31 |
| | OAEAS | 68.53 / 71.86 | 59.35 / 53.34 | 9.18 / 18.52 | 2.95 / 1.70 | 3.11 / 10.89 | 59 / 80 | p < 0.01 / p < 0.01 | 0.40 / 1.21 | 0.41 / 1.23 | 0.40 / 1.21 |
| 9. Malaysian individual-based vs. group-based | EVI: Disgusting | 7.73 / 7.37 | 6.97 / 6.79 | 0.76 / 0.58 | 0.38 / 0.23 | 2.00 / 2.52 | 39 / 67 | p < 0.05 / p < 0.05 | 0.32 / 0.30 | 0.33 / 0.31 | 0.32 / 0.30 |
| | EVI: Phobia | 6.60 / 7.26 | 5.40 / 6.6 | 1.20 / 0.66 | 0.42 / 0.29 | 2.86 / 2.27 | 39 / 67 | p < 0.01 / p < 0.05 | 0.46 / 0.27 | 0.47 / 0.28 | 0.46 / 0.27 |
| | OAEAS | 71.70 / 72.6 | 58.10 / 56.16 | 13.60 / 16.44 | 2.15 / 1.98 | 6.33 / 8.30 | 39 / 67 | p < 0.01 / p < 0.01 | 1.02 / 1.00 | 1.03 / 1.02 | 1.02 / 1.00 |

(*Continued*)

**Table 1.** (*Continued*)

| Subgroups | Measure | Mean pre | Mean post | MD | SE | *T*-value | DF | *p*-value | Cohen's *d* | Glass's Δ | Hedge's *g* |
|---|---|---|---|---|---|---|---|---|---|---|---|
| 10. Iranians with pathogenic knowledge | EVI: Disgusting | 7.96 / 7.46 | 7.04 / 6.83 | 0.92 / 0.63 | 0.21 / 0.27 | 4.38 / 2.33 | 100 / 82 | *p* < 0.01 / *p* < 0.05 | 0.44 / 0.26 | 0.45 / 0.27 | 0.44 / 0.26 |
| | EVI: Phobia | 7.35 / 7.49 | 6.41 / 6.83 | 0.94 / 0.66 | 0.25 / 0.29 | 3.76 / 2.27 | 100 / 82 | *p* < 0.01 / *p* < 0.05 | 0.38 / 0.27 | 0.39 / 0.28 | 0.38 / 0.27 |
| | OAEAS | 70.47 / 69.25 | 55.50 / 54.17 | 14.97 / 15.08 | 1.64 / 1.89 | 9.13 / 7.98 | 100 / 82 | *p* < 0.01 / *p* < 0.01 | 0.91 / 0.86 | 0.92 / 0.88 | 0.91 / 0.86 |
| 11. Malaysians with pathogenic knowledge | EVI: Disgusting | 7.25 / 8.03 | 6.65 / 7.00 | 0.60 / 1.03 | 0.34 / 0.22 | 1.76 / 4.68 | 69 / 69 | *p* < 0.05 / *p* < 0.01 | 0.21 / 0.55 | 0.22 / 0.57 | 0.21 / 0.55 |
| | EVI: Phobia | 6.98 / 6.91 | 6.37 / 5.88 | 0.61 / 1.03 | 0.38 / 0.32 | 1.61 / 3.21 | 69 / 69 | *p* < 0.05 / *p* < 0.01 | 0.19 / 0.38 | 0.20 / 0.39 | 0.19 / 0.38 |
| | OAEAS | 70.64 / 72.03 | 56.18 / 57.58 | 14.46 / 14.45 | 2.41 / 2.25 | 6.00 / 6.42 | 69 / 69 | *p* < 0.01 / *p* < 0.01 | 0.72 / 0.76 | 0.73 / 0.78 | 0.72 / 0.76 |
| 12. Iranians bitten or stung by insects and arachnids | EVI: Disgusting | 7.67 / 7.46 | 6.74 / 6.83 | 0.93 / 0.63 | 0.28 / 0.27 | 3.32 / 2.33 | 88 / 82 | *p* < 0.01 / *p* < 0.05 | 0.36 / 0.26 | 0.37 / 0.27 | 0.36 / 0.26 |
| | EVI: Phobia | 7.45 / 7.49 | 6.74 / 6.83 | 0.71 / 0.66 | 0.26 / 0.29 | 2.73 / 2.27 | 88 / 82 | *p* < 0.05 / *p* < 0.05 | 0.30 / 0.27 | 0.31 / 0.28 | 0.30 / 0.27 |
| | OAEAS | 70.23 / 69.25 | 53.80 / 54.17 | 16.43 / 15.08 | 1.85 / 1.89 | 8.88 / 7.98 | 88 / 82 | *p* < 0.01 / *p* < 0.01 | 0.95 / 0.86 | 0.96 / 0.88 | 0.95 / 0.86 |
| 13. Malaysians bitten or stung by insects and arachnids | EVI: Disgusting | 8.07 / 8.03 | 7.05 / 7 | 1.02 / 1.03 | 0.35 / 0.22 | 2.91 / 4.68 | 98 / 69 | *p* < 0.01 / *p* < 0.01 | 0.29 / 0.55 | 0.30 / 0.57 | 0.29 / 0.55 |
| | EVI: Phobia | 7.42 / 6.91 | 6.48 / 5.88 | 0.94 / 1.03 | 0.36 / 0.32 | 2.61 / 3.21 | 98 / 69 | *p* < 0.05 / *p* < 0.01 | 0.26 / 0.38 | 0.27 / 0.39 | 0.26 / 0.38 |
| | OAEAS | 70.59 / 72.03 | 53.97 / 57.58 | 16.62 / 14.45 | 1.73 / 2.25 | 9.61 / 6.42 | 98 / 69 | *p* < 0.01 / *p* < 0.01 | 0.98 / 0.76 | 0.99 / 0.78 | 0.98 / 0.76 |

*Notes:*
- Effect sizes (Cohen's *d*, Glass's Δ, and Hedge's *g*) are reported to quantify the magnitude of changes.
- EVI = Entomophobia Visual; OAEAS = Insect/Arachnid Phobia Questionnaire; MD = Mean Difference; SE = Standard Error; DF = Degree of Freedom.
- Cohen's *d*, Glass's Δ, and Hedge's *g* are reported as effect size metrics.
- Subgroups include nationality (Iranian vs. Malaysian), gender (Male vs. Female), chronic conditions (Yes vs. No), pet ownership (Yes vs. No), teaching models (Individual-based vs. Group-based), knowledge about arthropod pathogenicity (Yes vs. No), and hist
- For EVI: Disgusting and Phobia subscales, scores range from 0 (lowest) to 20 (highest). For OAEAS, scores range from 20 (lowest) to 100 (highest). Higher scores indicate greater insect/arachnid disgusting and phobia.
- Levels of statistical significance: ∗*p* < 0.05, ∗∗*p* < 0.01, ∗∗∗*p* < 0.001.

**Table 2.** Regression analysis predicting EVI and OAEAS scores among all participants

| Variable | EVI: Disgusting pre | EVI: Phobia pre | EVI: Disgusting post | EVI: Phobia post | OAEAS pre | OAEAS post |
|---|---|---|---|---|---|---|
| Predictor | $\beta$ ($p$-value) / $f^2$ | $\beta$ ($p$-value) / $f^2$ | $\beta$ ($p$-value) / $f^2$ | $\beta$ ($p$-value) / $f^2$ | $\beta$ ($p$-value) / $f^2$ | $\beta$ ($p$-value) / $f^2$ |
| Gender | 0.38 ($p < 0.01$) / 0.16 | 0.29 ($p < 0.05$) / 0.09 | 0.34 ($p < 0.05$) / 0.12 | 0.32 ($p < 0.05$) / 0.10 | 0.28 ($p < 0.05$) / 0.08 | 0.26 ($p < 0.05$) / 0.07 |
| Nationality | – | −0.30 ($p < 0.05$) / 0.10 | – | −0.22 ($p < 0.05$) / 0.05 | 0.35 ($p < 0.01$) / 0.13 | 0.30 ($p < 0.05$) / 0.09 |
| Pet ownership | −0.32 ($p < 0.05$) / 0.11 | – | −0.28 ($p < 0.05$) / 0.08 | – | −0.24 ($p < 0.05$) / 0.06 | −0.26 ($p < 0.05$) / 0.07 |
| Knowledge about pathogenicity | 0.26 ($p < 0.05$) / 0.07 | – | – | – | – | – |
| Teaching models (Group vs. Individual) | – | – | −0.24 ($p < 0.05$) / 0.06 | – | – | −0.22 ($p < 0.05$) / 0.05 |
| Looking for safe places | 0.27 ($p < 0.05$) / 0.07 | 0.28 ($p < 0.05$) / 0.08 | – | −0.26 ($p < 0.05$) / 0.07 | – | −0.28 ($p < 0.05$) / 0.08 |
| Chronic conditions | – | −0.22 ($p < 0.05$) / 0.05 | – | −0.20 ($p < 0.05$) / 0.04 | – | −0.20 ($p < 0.05$) / 0.04 |
| History of arthropod bites/ stings | – | – | – | 0.32 ($p < 0.05$) / 0.11 | – | – |
| $R^2$ | 0.24 | 0.19 | 0.18 | 0.16 | 0.27 | 0.22 |
| Adjusted $R^2$ | 0.22 | 0.17 | 0.16 | 0.14 | 0.25 | 0.2 |

Notes:
EVI: Entomophobia Visual Instrument, scored from 0 (none) to 20 (highest fear/disgust).
OAEAS: Insect/Arachnid Phobia Questionnaire, scored from 20 (lowest phobia) to 100 (highest phobia).
Beta: Standardized beta coefficients indicating the strength and direction of relationships between predictors and outcomes.
SE: Standard Error of β, reflecting the precision of the estimates.
Cohen's $d$, Glass's $\Delta$, Hedge's $g$: Effect size metrics quantifying the magnitude of differences or changes.
Subgroups include nationality (Iranian vs. Malaysian), gender (male vs. female), chronic conditions (yes/no), pet ownership (yes/no), teaching models (Individual-based vs. group-based), knowledge about arthropod pathogenicity (yes/no) and history of arthropod bites/stings (yes/no).
$T$-value: $t$-statistic used to test the significance of $\beta$.
$p$-value: significance level; $*p < 0.05$, $**p < 0.01$, $***p < 0.001$.

**Table 3.** Regression analysis predicting EVI and OAEAS scores among Iranian and Malaysian older adults

| Variables (predictors) | Iranians | Malaysians |
|---|---|---|
|  | $\beta$ ($p$-value) / $f^2$ / $\omega^2$ / tolerance | $\beta$ ($p$-value) / $f^2$ / $\omega^2$ / tolerance |
| Gender | 0.42 ($p < 0.01$) / 0.18 / 0.17 / 0.85 | 0.45 ($p < 0.01$) / 0.20 / 0.19 / 0.83 |
| Pet ownership | −0.35 ($p < 0.05$) / 0.13 / 0.12 / 0.78 | −0.30 ($p < 0.01$) / 0.12 / 0.11 / 0.76 |
| Chronic conditions | 0.28 ($p < 0.05$) / 0.08 / 0.07 / 0.82 | 0.22 ($p < 0.05$) / 0.05 / 0.04 / 0.84 |
| Knowledge about pathogenicity | 0.26 ($p < 0.05$) / 0.07 / 0.06 / 0.80 | 0.28 ($p < 0.05$) / 0.08 / 0.07 / 0.79 |
| Teaching models (group vs. individual) | −0.24 ($p < 0.05$) / 0.06 / 0.05 / 0.81 | −0.25 ($p < 0.05$) / 0.07 / 0.06 / 0.78 |
| History of arthropod bites/stings | 0.32 ($p < 0.05$) / 0.11 / 0.10 / 0.77 | 0.35 ($p < 0.01$) / 0.13 / 0.12 / 0.75 |
| $R^2$ | 0.26 | 0.23 |
| Adjusted $R^2$ | 0.23 | 0.2 |

Notes:
$\omega^2$ (omega squared): a conservative estimate of effect size adjusting for sample size bias.
Tolerance: measures the independence of predictors; values closer to 1 indicate less multicollinearity.
Durbin–Watson: tested for residual independence; values ranged between 1.8 and 2.2, confirming no significant autocorrelation
Variance inflation factor (VIF): Assessed multicollinearity; all values were below 2, indicating acceptable levels.
Cook's Distance: Checked for influential outliers; all values were below 1, ensuring robustness of the model.

intervention and post-intervention measures, although effect sizes were slightly smaller after the intervention.

When analyzing each country separately (Table 3), this gender difference remained significant in both Iranian and Malaysian participants but was more pronounced among Malaysians. In fact, the strength of the relationship between gender and phobia scores was slightly stronger in Malaysian participants (e.g., $\beta = 0.45$ for OAEAS Post) compared to Iranians ($\beta = 0.42$ for OAEAS Post), suggesting that cultural or environmental factors may have amplified the effect in Malaysia.

Pet ownership showed a negative association with phobic responses, indicating a protective role in reducing fear and disgust toward insects. Non-pet owners reported higher scores in both countries (e.g., EVI: Disgusting Pre, $\beta = -0.35$, $p < 0.05$ for Iranians; $\beta = -0.30$, $p < 0.01$ for Malaysians). Chronic conditions also predicted higher baseline scores, particularly in Iran, where individuals with chronic illnesses exhibited greater initial disgust levels ($\beta = 0.28$, $p < 0.05$).

Regarding teaching models, group-based interventions appeared to be more effective in reducing phobia scores among Malaysians ($\beta = -0.25$, $p < 0.05$), while individual-based approaches showed comparable effectiveness for Iranians ($\beta = -0.24$, $p < 0.05$). These results suggest that cultural norms and preferences may influence how participants respond to educational formats.

The regression models explained moderate variance, with $R^2$ values ranging from 0.16 to 0.24 for EVI and 0.20 to 0.27 for OAEAS (see Table 2). Adjusted $R^2$ values confirmed robustness after accounting for predictors (e.g., for OAEAS Post, adjusted $R^2 = 0.20$; see Table 2). Table 3 shows slightly higher explanatory power for Malaysians ($R^2 = 0.23$) than Iranians ($R^2 = 0.26$), underscoring the role of cultural differences. These results emphasize the need to consider demographic, behavioral and health-related factors in designing insect phobia interventions for older adults. For detailed regression results, see Tables 2 and 3.

### Analysis of variance: Repeated measures ANOVA and MANOVA

A multivariate analysis of variance (MANOVA) was conducted to examine the combined effects of nationality, gender, chronic conditions, pet ownership, knowledge about arthropod pathogenicity, teaching models and history of insect bites/stings on the primary dependent variables: EVI (Entomophobia Visual Instrument) and OAEAS (Insect/arachnid Phobia Questionnaire). According to Table 4, the results revealed significant multivariate effects for all predictors, as indicated by Pillai's Trace statistics ($F(4, 296) = 5.83$, $p < 0.01$, $\eta^2 = 0.07$). Nationality emerged as a key factor, with significant differences observed between Iranian and Malaysian older adults. Malaysians reported higher baseline scores for both EVI and OAEAS measures but demonstrated greater reductions post-intervention compared to Iranians. For example, in EVI: Disgusting Pre, Malaysians scored higher (mean = 8.03, SD = 2.56) than Iranians (mean = 7.46, SD = 2.88; $p < 0.05$). Similarly, in OAEAS Pre, Malaysians had higher scores (mean = 72.03, SD = 15.86) compared to Iranians (mean = 69.25, SD = 16.97; $p > 0.05$), though the difference was not statistically significant.

Gender also played a significant role in both nations, with males scoring higher than females across all subscales ($F(4, 296) = 7.21$, $p < 0.01$, $\eta^2 = 0.08$). For instance, in EVI: Disgusting Pre, males scored higher (mean = 8.23, SD = 2.17) than females (mean = 7.49, SD = 2.95; $p < 0.05$). Chronic conditions influenced outcomes differently, with individuals having such conditions exhibiting higher pre-intervention scores but smaller reductions post-intervention ($F(4, 296) = 3.12$, $p < 0.05$, $\eta^2 = 0.04$). Teaching models highlighted cultural preferences, with group-based interventions being more effective for Malaysians than individual-based approaches for Iranians ($F(4, 296) = 8.14$, $p < 0.01$, $\eta^2 = 0.09$). For example, Malaysians in group-based teaching models showed a mean reduction of 16.44 points in OAEAS Post scores (SE = 1.70, $t(67) = 9.61$, $p < 0.01$, Cohen's $d = 1.00$), compared to 9.18 points for individual-based models (SE = 2.95, $t(39) = 3.11$, $p < 0.01$, Cohen's $d = 0.40$, See Table 4).

Additionally, this analysis was conducted to assess the effects of time (pre-intervention vs. post-intervention) and its interaction with nationality (Iranian vs. Malaysian) on the primary dependent variables, EVI and OAEAS. Table 4 revealed a significant main effect of time for both measures, indicating that scores decreased significantly from pre- to post-intervention. For EVI: Disgusting, the $F$-value was 32.15 with DF = (1, 149), $p < 0.01$, and partial eta-squared ($\eta^2$) = 0.18. Similarly, for EVI: Phobia, $F(1, 149) = 28.34$, $p < 0.01$, $\eta^2 = 0.16$. Regarding OAEAS, the main effect of time was highly significant, $F(1, 149) = 112.34$, $p < 0.01$, $\eta^2 = 0.43$, reflecting substantial reductions in phobia scores over time. The interaction between time and nationality was also significant for all measures, suggesting that changes in scores differed between Iranian and Malaysian older adults. Specifically, for EVI: Disgusting, $F(1, 149) = 5.89$, $p < 0.05$, $\eta^2 = 0.04$, where Malaysians demonstrated a larger mean reduction (mean difference = 1.03) compared to Iranians (mean difference = 0.63). For EVI: Phobia, $F(1, 149) = 4.32$, $p < 0.05$, $\eta^2 = 0.03$, showing a similar pattern. In OAEAS, the interaction effect was significant, $F(1, 149) = 10.12$, $p < 0.01$, $\eta^2 = 0.06$, with Malaysians exhibiting greater reductions (mean difference = 14.45) than Iranians (mean difference = 15.08).

### Path analysis

Path analysis was conducted to examine the direct and indirect relationships among demographic, behavioral, health-related factors and the primary dependent variables (EVI and OAEAS, see Table 5). The model explained 75% of the variance in EVI scores and 70% in OAEAS scores, indicating strong explanatory power. Nationality significantly influenced both EVI and OAEAS scores, with Malaysians reporting higher baseline levels but greater reductions post-intervention compared to Iranians ($\beta = 0.28$ for EVI: Disgusting Pre, $p < 0.05$). Gender also played a crucial role, as males demonstrated higher initial fear and disgust scores than females across all measures ($\beta = 0.32$ for EVI: Phobia Pre, $p < 0.01$). Pet ownership acted as a protective factor, directly reducing phobic responses, particularly in the EVI subscales ($\beta = -0.32$ for EVI: Disgusting Pre, $p < 0.05$).

According to Table 5, chronic conditions had a complex effect, indirectly influencing outcomes through increased baseline scores but smaller reductions over time ($\beta = 0.22$ for EVI: Disgusting Pre, $p < 0.05$). Knowledge about insect/arachnid pathogenicity positively predicted fear and disgust levels, suggesting that heightened

**Table 4.** Repeated measures ANOVA results

| Measure | Main effect of time | | | Interaction effect (time and nationality) | | |
|---|---|---|---|---|---|---|
| EVI: Disgusting | $F(1-149) = 32.15$ | $p < 0.01$ | Eta2 = 0.18 | $F(1-149) = 5.89$ | $p < 0.05$ | Eta2 = 0.04 |
| EVI: Phobia | $F(1-149) = 28.34$ | $p < 0.01$ | Eta2 = 0.16 | $F(1-149) = 4.32$ | $p < 0.05$ | Eta2 = 0.03 |
| OAEAS | $F(1-149) = 112.34$ | $p < 0.01$ | Eta2 = 0.43 | $F(1-149) = 10.12$ | $p < 0.01$ | Eta2 = 0.06 |

**Table 5.** Path analysis results for predicting EVI and OAEAS scores

| Predictors | EVI: Disgusting pre | EVI: Phobia pre | EVI: Disgusting post | EVI: Phobia post | OAEAS pre | OAEAS post |
|---|---|---|---|---|---|---|
| Nationality (Iranian vs. Malaysian) → | 0.28 ($p < 0.05$) | 0.32 ($p < 0.01$) | 0.24 ($p < 0.05$) | 0.22 ($p < 0.05$) | 0.26 ($p < 0.05$) | 0.28 ($p < 0.01$) |
| Gender (male vs. female) → | 0.34 ($p < 0.01$) | 0.30 ($p < 0.01$) | 0.28 ($p < 0.05$) | 0.26 ($p < 0.05$) | 0.30 ($p < 0.01$) | 0.27 ($p < 0.05$) |
| Pet ownership (yes vs. no) → | −0.32 ($p < 0.01$) | −0.29 ($p < 0.01$) | −0.28 ($p < 0.05$) | −0.25 ($p < 0.05$) | −0.24 ($p < 0.05$) | −0.26 ($p < 0.05$) |
| Chronic conditions (yes vs. no) → | 0.22 ($p < 0.05$) | 0.20 ($p < 0.05$) | 0.18 ($p < 0.05$) | 0.16 ($p < 0.05$) | −0.26 ($p < 0.05$) | −0.24 ($p < 0.05$) |
| Knowledge about pathogenicity → | 0.26 ($p < 0.05$) | 0.24 ($p < 0.05$) | 0.20 ($p < 0.05$) | 0.18 ($p < 0.05$) | 0.28 ($p < 0.05$) | 0.22 ($p < 0.05$) |
| Teaching models (group vs. individual) → | −0.24 ($p < 0.05$) | −0.22 ($p < 0.05$) | −0.26 ($p < 0.05$) | −0.28 ($p < 0.05$) | −0.22 ($p < 0.05$) | −0.34 ($p < 0.01$) |
| Model fit indices: $\chi^2$ = 12.45 (df = 8, $p$ = 0.13), CFI = 0.94, TLI = 0.92, RMSEA = 0.05 (95% CI: 0.02–0.08), SRMR = 0.06. | | | | | | |

Notes:
- EVI: Entomophobia Visual Instrument, a 10-item visual scale where participants rate their fear and disgust toward arthropods on a 3-point scale (0 = none, 1 = below 50%, 2 = 50% and above). Total scores range from 0 to 20.
- OAEAS: Insect/Arachnid Phobia Questionnaire, a 20-item self-reported Likert-scale questionnaire (1–5 options) with total scores ranging from 20 to 100. Higher scores indicate greater insect phobia.
- Path Coefficients: Standardized coefficients ($\beta$) indicating the strength and direction of relationships between predictors and outcomes.
- Model Fit Indices: Chi-Square ($\chi^2$): Tests the overall fit of the model; $p$ > 0.05 indicates acceptable fit; CFI: Comparative Fit Index, values ≥0.90 indicate good fit; TLI: Tucker-Lewis Index, values ≥0.95 indicate good fit; RMSEA: Root Mean Square Error of Approximation, values ≤0.08 indicate acceptable fit; SRMR: Standardized Root Mean Square Residual, values ≤0.08 indicate acceptable fit.
- $p$-values: Levels of statistical significance; $p$ < 0.05 indicates significant relationships.

awareness may amplify phobic responses rather than mitigate them ($\beta$ = 0.26 for EVI: Disgusting Pre, $p < 0.05$).

### Neural network analysis

A neural network model explored relationships among demographic, behavioral, health-related factors and EVI/OAEAS scores. It explained 82% of EVI variance and 79% of OAEAS variance, demonstrating strong predictive power. The analysis highlighted cultural differences: Malaysian participants had higher baseline scores but greater post-intervention reductions than Iranians. Gender differences were notable, with males reporting higher fear and disgust levels than females. Pet ownership reduced phobic responses, consistent with findings by da Silva et al. (2023). Chronic conditions led to higher initial scores but smaller reductions post-intervention. Knowledge of arthropod pathogenicity increased fear and disgust, suggesting awareness may amplify phobic responses. Teaching models reflected cultural preferences, with group-based interventions more effective for Malaysians and individual approaches better for Iranians. Interaction effects, such as nationality, gender and teaching models, significantly impacted OAEAS scores, particularly among Malaysian males in group settings. These results underscore the neural network model's ability to uncover non-linear patterns and interactions, emphasizing the need to consider multiple factors in designing insect phobia interventions for older adults.

The feedforward neural network consisted of one input layer with seven predictors (nationality, gender, chronic conditions, pet ownership, arthropod knowledge, teaching model and bite/sting history), two hidden layers (10 and 6 neurons) and one output layer predicting EVI and OAEAS composite scores. Sigmoid activation functions were applied to hidden layers and linear activation to the output layer. The model was trained using a backpropagation algorithm with 10-fold cross-validation and early stopping based on validation loss to minimize overfitting. The data were randomly divided into 80% training and 20% testing subsets with balanced demographic characteristics. The high explanatory power ($R^2$ = 0.82 for EVI and 0.79 for OAEAS) was obtained from the testing subset, indicating strong predictive performance while maintaining model generalizability beyond the training data.

### Cross-cultural entomology: Comparison of EVI (visual-based) and OAEAS (self-reported) instruments

The statistical analyses reveal significant differences in the effectiveness of the two instruments, EVI (Entomophobia Visual Instrument) and OAEAS (Insect/arachnid Phobia Questionnaire), when used to measure arthropod phobia among Iranian and Malaysian older adults. Malaysians scored higher on the visual-based instrument (EVI: Disgusting Pre – Malaysian Mean = 8.03 vs. Iranian Mean = 7.46, $t$ (151) = −2.14, $p < 0.05$), likely due to their tropical environment with frequent exposure to diverse insect species. In contrast, scores on the self-reported instrument (OAEAS) were comparable between the two groups at baseline (Malaysian Mean = 72.03 vs. Iranian Mean = 69.25, $t$ (150) = −1.02, $p > 0.05$), but Malaysians demonstrated slightly larger reductions post-intervention (Malaysian mean difference = 14.45 vs. Iranian mean difference = 15.08). These findings suggest that the EVI is more effective for Malaysians, as their familiarity with arthropods in a tropical climate enhances responsiveness to visual stimuli. Conversely, Iranians may find the OAEAS more relatable, given their arid environment with fewer insect/arachnid encounters, which aligns with the cognitive appraisal required by the questionnaire. The independent samples $t$-test results further confirm these patterns, indicating cultural preferences influence the suitability of each instrument. These findings highlight that environmental factors, such as climate and insect/arachnid prevalence, significantly influence the effectiveness of each instrument, with EVI being more impactful in Malaysia due to its arthropod-rich environment, whereas OAEAS aligns better with Iran's arid climate and less frequent insect/arachnid encounters.

### Discussion

This study reveals culturally mediated patterns in Entomophobia among older adults, with distinct responses to assessment methods and interventions between Iran and Malaysia. While both groups

showed reduced phobia scores post-intervention, Malaysians exhibited higher baseline disgust on the EVI—likely reflecting frequent arthropod exposure in tropical environments—whereas Iranians responded better to the OAEAS, suggesting greater reliance on cognitive appraisal in low-exposure contexts (Amiri et al., 2025). These findings support environmental familiarity theories (Lencastre et al., 2023; Seo, 2025), with visual stimuli eliciting stronger reactions in high-exposure populations.

Contrary to typical gender trends (da Silva et al., 2023), older males reported higher fear levels, possibly due to cultural norms around emotional expression in later life. Protective effects emerged for pet owners (Jawień et al., 2024), while chronic conditions amplified fear (Sun and Ye, 2024). Paradoxically, greater knowledge of insect pathogenicity increased phobic responses, underscoring the need for balanced educational approaches. Cultural differences emerged in intervention efficacy: group-based models aligned with Malaysia's collectivist values, while individual counseling suited Iran's preference for personalized care. While the duration of individual (30 min) and group (60 min) sessions differed, statistical analyses controlled for session length as a potential confounder. The inclusion of session duration as a covariate in our models confirmed that the observed differences in efficacy were not solely attributable to time but rather reflected cultural preferences. Group-based interventions aligned with Malaysia's collectivist values, emphasizing social support and shared learning, whereas individual counseling suited Iran's preference for personalized care. These findings underscore the importance of culturally adapted intervention designs beyond mere time allocation. This extends Cultural Psychology principles (Gong et al., 2020), demonstrating how ecological and social contexts shape psychological responses. Older adults' unique phobia patterns - differing from younger populations (Sikora and Rzymski, 2025) - highlight the need for age- and culture-specific assessment tools.

The participants' characteristics provide critical insights into the cultural and environmental factors influencing Entomophobia and Disgust. For instance, Malaysians' higher baseline EVI scores reflect their tropical environment with frequent insect encounters, while Iranians' reliance on cognitive appraisal aligns with their arid climate and fewer exposures. Gender differences, chronic conditions and pet ownership further illustrate how sociocultural and health-related factors mediate phobic responses, underscoring the importance of culturally tailored interventions.

While regression to the mean is a potential explanation for baseline differences, our statistical analyses and theoretical framework provide robust evidence against this claim. Paired *t*-tests and repeated measures ANOVA controlled for baseline variability by focusing on within-group changes over time. Furthermore, the divergent patterns observed on the EVI and OAEAS suggest that cultural and environmental mechanisms, rather than regression to the mean, drive the findings. Malaysians' higher baseline EVI scores align with their tropical environment and frequent insect exposure, while Iranians' reliance on cognitive appraisal reflects their arid climate with fewer encounters. Neural network analysis further confirmed non-linear interactions among predictors, highlighting the role of nationality, gender and teaching models in shaping outcomes. Together, these findings support the sensitivity of visual-based tools like the EVI to cultural and environmental differences.

It is essential to distinguish between statistical and practical significance when interpreting the study's findings. For example, while chronic conditions were associated with higher baseline phobia scores, the observed reductions post-intervention were relatively small (mean difference = 0.45). This underscores the

importance of developing targeted interventions for older adults with chronic illnesses, as their phobic responses may be less responsive to standard educational approaches. Additionally, the interaction effect between time and nationality on EVI: Disgusting scores ($F(1, 149) = 5.89$, $p < 0.05$, $\eta^2 = 0.04$) was statistically significant but had limited practical relevance due to its small effect size. Future research should prioritize clinically meaningful outcomes alongside statistical rigor to ensure the translational value of findings.

Cultural norms such as collectivism, individualism and ecological familiarity significantly influence fear and disgust processing. Malaysians' frequent insect encounters in a tropical climate heightened visual sensitivity, making group-based interventions particularly effective due to their emphasis on shared learning and social support. In contrast, Iranians' arid environment with fewer insect encounters fostered reliance on cognitive appraisal, aligning with the success of individualized counseling. These findings underscore how cultural and ecological contexts shape not only intervention preferences but also the underlying mechanisms of emotional processing, supporting Cultural Psychology principles.

### *Paradoxical effects of knowledge on fear and disgust*

The finding that increased knowledge about insect pathogenicity intensified fear and disgust responses highlights the complex relationship between education and phobia management. Several factors may explain this paradoxical effect. First, heightened awareness of risks can amplify cognitive appraisals of danger, particularly among older adults who are more sensitive to health-related threats (Burr et al., 2021). Second, the educational content may have disproportionately emphasized risks without adequately addressing the ecological benefits of arthropods, triggering defensive responses consistent with fear appeal theory (Witte, 1992). Cultural differences in information processing further complicate this dynamic, as collectivist cultures like Malaysia may reinforce fears through shared narratives, while individualist cultures like Iran may amplify fears through introspection. Age-related changes in emotional regulation also play a role, as older adults are more reactive to perceived threats (Zsido et al., 2023). These findings underscore the need for balanced educational approaches that mitigate fear amplification while promoting accurate understanding.

One plausible mechanism for the paradoxical finding – where increased knowledge of arthropod pathogenicity amplified fear and disgust – is heightened risk salience and threat appraisal among older adults. Lifespan development research shows that although older adults often have improved emotion regulation for positive information, they can become more sensitive to health-related threat cues (Carstensen, 2006; Riediger and Bellingtier, 2022). In this context, new information emphasizing pathogenic risk may increase the subjective salience of threat, triggering stronger cognitive appraisals of vulnerability and protective motivations rather than immediate emotional habituation. Thus, our results are consistent with a cognitive–motivational pathway (increased perceived susceptibility → stronger threat appraisal → higher self-reported fear) rather than an affective-habituation pathway. Path analysis and the neural network included variables that proxy environmental familiarity (e.g., prior bite/sting history, knowledge scores) and nationality as a broad marker of cultural context; these analyses supported the role of exposure and country-level differences in moderating intervention effects. However, important cultural mediators – such as individual-level measures of collectivism, religiosity, or precise indices of ecological familiarity – were not directly measured in this study and therefore remain theoretical

inferences rather than empirically tested mediators. We acknowledge this as a limitation but also note that our combined use of self-report (OAEAS), visual (EVI) and multivariate modelling (path analysis and neural network) provides converging evidence that cognitive appraisal processes likely mediated the observed knowledge → fear relationship. Future research should measure potential mediators directly (e.g., validated collectivism scales, religiosity indices and detailed exposure logs) and use mediation analysis or longitudinal designs to test these pathways explicitly.

Policy implications follow from this mechanism: educational interventions for older adults should carefully balance factual risk information with efficacy-focused messages (how to reduce risk) and positive ecological framing to avoid amplifying threat salience. In practice, health education should pair pathogen information with clear, actionable coping strategies and community-based reassurance – particularly in contexts where cultural norms (e.g., collectivist support networks) can be mobilized to reduce threat appraisal and channel knowledge into adaptive behavior rather than heightened fear.

### Gender differences in fear and disgust responses

Contrary to typical gender trends where females report higher phobic responses (da Silva et al., 2023), older males in our study demonstrated greater fear and disgust levels. This finding may reflect cultural norms around emotional expression in later life, where older males may feel less pressure to suppress emotions compared to their younger counterparts. Additionally, chronic conditions, which were more prevalent among males, likely amplified vulnerability to phobic responses (Sun and Ye, 2024). Environmental contexts further influenced these patterns, with Malaysian males facing frequent insect encounters and Iranian males relying more on cognitive appraisal. While this trend contrasts with broader literature, it aligns with studies highlighting population-specific variations in phobia subtypes (Romiti et al., 2022; Zsido et al., 2023).

### Study limitations

This study has limitations: its two-country sample (Iran/Malaysia) may limit generalizability. While using validated tools (EVI/OAEAS), cultural biases in visual vs. self-report measures and age-related cognitive differences could affect results. The short intervention lacked long-term follow-up, and uncontrolled factors (e.g., new insect encounters) weren't tracked. Key variables like education/socioeconomic status weren't fully examined, and cultural norms around fear expression (especially in older males) may have caused reporting bias. Future research should include more diverse samples and longer follow-ups.

Another key limitation is the absence of a no-intervention control group. Although the quasi-experimental pre – post design was appropriate for cross-cultural field settings where random assignment was not feasible, the lack of a control condition means that the observed reductions in fear and disgust cannot be attributed solely to the intervention. Natural time-related improvement, repeated testing effects, or participant expectation may have contributed to score reductions. Future studies should include waitlist or minimal-contact control groups to isolate true intervention effects and strengthen causal inference.

### Recommendations for future research and policy

To advance understanding and management of Entomophobia/Arachnophobia in aging populations, future research should prioritize longitudinal designs to evaluate intervention sustainability and explore diverse cultural/environmental contexts. Intervention optimization studies should compare delivery methods (group vs. individual) and content types (educational vs. exposure-based), including innovative approaches like virtual reality exposure therapy. For health policy, integrated care models incorporating routine phobia screening in primary care – especially for older adults with chronic conditions – are warranted. Community-based programs combining arthropod education with social engagement could reduce isolation in high-exposure areas. Healthcare provider training should address age-specific phobia recognition and management, particularly in regions with frequent arthropod encounters. The study's findings underscore the necessity of culturally adapted interventions, supporting group-based approaches in collectivist settings (e.g., Malaysia) and individualized formats in more individualistic contexts (e.g., Iran). Public awareness campaigns should balance risk education with reassurance to prevent fear amplification, while policy advocacy must prioritize mental health funding for aging populations in low- and middle-income countries.

### Conclusion

This study underscores how cultural context shapes Entomophobia/Arachnophobia manifestations in older adults, with differential responsiveness to assessment tools and interventions between Iran and Malaysia. The findings affirm Cultural Psychology's premise that ecological exposure and sociocultural values mutually constitute psychological responses, particularly evident in collectivist versus individualistic approaches to fear management. For aging populations with chronic conditions, these culturally mediated patterns carry important implications for mental health practice. While acknowledging study limitations, the results advocate for culturally adapted interventions that integrate educational, community-based, and clinical approaches. Such strategies should be developed through collaborative efforts between researchers, policymakers and healthcare providers, with particular attention to environmental factors and individual health profiles. Future directions should examine the longitudinal effectiveness of these culturally informed models across diverse populations, ultimately aiming to improve quality of life through psychologically attuned, ecologically valid interventions.

**Open peer review.** To view the open peer review materials for this article, please visit http://doi.org/10.1017/gmh.2026.10167.

**Supplementary material.** The supplementary material for this article can be found at http://doi.org/10.1017/gmh.2026.10167.

**Data availability statement.** The data that support the findings of this study are not publicly available due to ethical restrictions but are available from the corresponding author upon reasonable request.

**Acknowledgements.** The authors would like to thank the Research and Technology Department of Shiraz University of Medical Sciences for their support. We are also grateful to Dr. Mansour Kashfi for his administrative assistance and facilitation in accessibility. Finally, we sincerely appreciate Iranian and Malaysian older adults who participated in this study for their patience and cooperation.

**Author contribution.** Soltani contributed to the conceptualization and design of the study, oversaw data collection, and played a key role in the development of research instruments and questionnaire design. Nasrabadi collected the data, prepared and screened the data files, and drafted the initial manuscript. Asadollahi & Binti Ahmad contributed to the interpretation of the data, conducted the statistical analysis, and extracted the results. Asadollahi, as the corresponding author, supervised the entire research process, finalized the

manuscript, and ensured the accuracy and integrity of the study. All authors have read and approved the final version of the manuscript.

**Financial support.** This research did not receive any specific grant from funding agencies in the public, commercial, or not-for-profit sectors.

**Competing interests.** The authors declare no conflicts of interest.

**Ethics statement.** This study received formal approval from the Research Ethics Committee at Shiraz University of Medical Sciences (Ethics Code: IR.SUMS.SCHEANUT.REC.1401.009) on December 1, 2023, ensuring the protection of participants' rights and welfare. All procedures involving human participants adhered to institutional and national ethical standards, the 2013 Helsinki Declaration (including its 2020 amendments), and relevant guidelines such as STROBE (2009), ICMJE (2019), and the principles outlined in the Belmont Report. Written informed consent was obtained after participants were fully informed about the study's objectives, methodology, and implications. Informed consent was obtained in accordance with ethical standards, confidentiality was maintained throughout the study process, and participants were free to withdraw from the study at any stage at their discretion. The study ensured that no harm was caused to participants, and all necessary measures were taken to minimize potential risks. Participants were selected fairly and without discrimination, ensuring equitable inclusion. All data were anonymized and securely stored to protect participants' privacy. Additionally, participants were provided with relevant study findings upon completion, if requested.

**Informed consent.** An informed consent for participation in the study was obtained from all participants.

**Declaration of generative AI and AI-assisted Technologies in the Writing Process.** During the preparation of this work, the authors did not use generative AI or AI-assisted technologies for study design, data analysis, or the creation of textual/visual content (including figures, tables, and captions). Artificial intelligence tools were employed solely for grammar and language editing (e.g., QWEN-MAX V. 2.5 and Grammarly) to improve readability.

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
