## [Reviewer Report]

This is a well-conceived study addressing an underexplored area at the intersection of gerontology, mental health, and cultural psychology. The authors investigate culturally adapted interventions for entomophobia and disgust among older adults in Iran and Malaysia using quasi-experimental methods.

The manuscript demonstrates strong analysis and cultural sensitivity, but requires greater theoretical depth, clearer explanations of methodology, and stronger interpretation of the data before it is ready for publication. This manuscript would also benefit from line-by-line editing for clarity in language usage and style. For example, “disgusting” is often used which should likely be changed to “disgust.” The following is a critique of each of the sections:

Introduction:

Page 3, Sentence starting on line 18 – This sentence doesn’t make sense to me as it is currently phrased.

Page 3, Paragraph starting at line 30 – again these sentences are confusing to me as they are currently phrased. They seem to be missing some words that would help clarify them.

The acronym of OAEAS (Older Adults’ Entomophobia & Arachnophobia Scale) is not defined until the method section but appears in the introduction and is unclear. Reference the scale after the first use of the acronym.

Overall, the introduction is too succinct and doesn’t make enough of a case for why this research was conducted. I think that the rationale for why this study was conducted could be strengthened.

The dual-nation comparison (Iran vs. Malaysia) adds valuable insight into how cultural and environmental contexts influence the effectiveness of behavioral interventions. However, the cultural analysis remains somewhat surface-level, primarily contrasting collectivist vs. individualist cultures without deeper engagement with cultural mechanisms (e.g., social learning, cognitive framing).

Method:

It is possible that the sessions varied on time rather than individual or group nature. Individual sessions were only 30 minutes while group session were 60 minutes. Is it possible that the differences in efficacy were not based on collectivist vs. individualist culture but on differences in the time spent on the topics?

Results:

I found the Participants’ Characteristics interesting, but don’t understand why those particular characteristics were chosen to be described and how this is important to the ultimate conclusions drawn. The idea that these highlight differences in the cultures as the authors conclude at the end of the section is unclear to me given that these differences do not seem to be discussed again.

From the paper: “Baseline scores for the Entomophobia Visual Instrument (EVI) and the Insect/arachnid Phobia Questionnaire (OAEAS) revealed significant cross-cultural differences. Malaysian participants scored higher on the EVI: Disgusting Pre (Mean = 8.03, SD = 2.56) compared to Iranians (Mean = 7.46, SD = 2.88; t(149) = -2.24, p < 0.05). However, no significant difference was observed in OAEAS Pre scores between Malaysians (Mean = 72.03, SD = 15.86) and Iranians (Mean = 69.25, SD = 16.97; t(147) = - 1.02, p > 0.05). These findings suggest that the visual-based EVI may be more sensitive to cultural and environmental differences, such as the higher prevalence of insects/arachnids in Malaysia.”

An alternative explanation that fits with this data is regression to the mean. On the EVI the Malaysian participants had a more extreme score at baseline, which upon second testing, was lessened not only because of the intervention but because of regression to the mean and this is the factor driving the entirety of the difference between the two cultures on the two measures. How can the authors defend against this claim?

The statistical presentation is detailed and includes effect sizes and fit indices, which is commendable. However, the tables in this paper are hard to interpret and it is likely that this paper would benefit from some figures rather than just tables. Or at least simplified tables. Further, practical significance should be distinguished from statistical significance — several effects (e.g., small mean differences) are statistically significant but may have limited clinical meaning.

Discussion:

The study aligns with cultural psychology but treats culture mainly as a contextual modifier.

The discussion could better articulate how cultural norms (e.g., collectivism, individualism, ecological familiarity) translate into behavioral differences in fear and disgust processing.

The paradoxical finding that increased knowledge about insect pathogenicity intensifies fear deserves more theoretical unpacking. Why is this the case? What are the possible explanations?

How can the authors explain the difference in typical gender trends beyond age-related changes in disclosure? Why would this finding be so consistent in other studies and not here?

Limitations acknowledged as well as the future directions are very well done.

---

## [Reviewer Report]

Thank you for allowing me to review this manuscript. The paper presents an innovative and methodologically sophisticated cross-cultural study examining entomophobia and disgust among older adults in Iran and Malaysia. The inclusion of neural network modeling and culturally tailored interventions is commendable. However, after carefully reading, I suggest revisions that are needed to improve conceptual clarity, methodological transparency, theoretical depth, and the articulation of cultural mechanisms.

Introduction

1. Clarify why these two countries were chosen? Is it primarily ecological (tropical vs. arid) or cultural (collectivist vs. individualist)?

2. Could you hypothesize which intervention would be more effective in each culture before conducting the study, based on prior literature?

Method

1. The study used a two-stage stratified random sampling method. Please provide more details on the two stages of stratification for both the Iranian and Malaysian samples to enhance replicability. For instance, what was the first stratification stage (e.g., city/district) before selecting individuals?

2. For EVI instrument uses a 3-point scale where "1 = below 50%“ and ”2 = 50% and above“. To my knowledge, this is an unusual response format for subjective rating scales. What exactly do ”below 50%“ and ”50% and above" refer to? Does it relate to the intensity of the subjective feeling (e.g., 50% intensity) or the likelihood of a behavioral response? Please provide clarity on the scale’s anchor points, is critical for interpreting the raw scores.

3. Were the same educators used in both cultural settings? If not, how were they standardized?

Results

1. The results section states: "OAEAS scores decreased more among Malaysians. Numerically, the mean difference for Iranians (15.08 points) is larger than for Malaysians (14.45 points). Please clarify this statement, confirm the correct mean differences, and state clearly which group had the greater reduction. Also, why did OAEAS scores show greater reductions than EVI in certain subgroups? Does this suggest cognitive desensitization rather than emotional habituation?

2. The quasi-experimental design is acceptable given the cross-cultural nature. However, the lack of a no-intervention control group means any pre-to-post change cannot be definitively attributed to the intervention over natural change or simple time effects. This should be explicitly discussed as a limitation.

3. I highly appreciate the use of a feedforward neural network model in this study. Please provide a brief explanation of the neural network’s architecture (e.g., number of hidden layers, nodes) and the cross-validation or generalization testing procedure to justify the high explanatory power. Further, given that the neural network explained 82% of the variance, how do you ensure model generalizability beyond the study sample?

Discussion

I proposed that more theoretical depth could be added to explain paradoxical findings (e.g., knowledge increasing fear). Could this be mediated by risk salience or threat appraisal in older adults? You can integrate lifespan development theory to contextualize older adults’ emotional regulation and discuss policy implications more concretely.

I wonder if there are any cultural mediators, such as collectivism, religiosity, or environmental familiarity, that have been statistically tested or inferred theoretically.

---

## [Reviewer Report]

The authors thoughtfully incorporated the reviewers’ feedback, resulting in a substantially strengthened manuscript. I am impressed by the manner in which they engaged with the critiques and used them to meaningfully enhance the overall quality of the work.